# Theoretical Computational Analysis Predicts Interaction Changes Due to Differences of a Single Molecule in DNA

Jun Koseki [1,*], Haruka Hirose [1], Masamitsu Konno [2] and Teppei Shimamura [1,*]

[1] Division of Systems Biology, Graduate School of Medicine, Nagoya University, Nagoya 466-8550, Japan
[2] Cellular and Molecular Biotechnology Research Institute, Advanced Industrial Science and Technology, Tokyo 135-0064, Japan
* Correspondence: jkoseki@med.nagoya-u.ac.jp (J.K.); shimamura@med.nagoya-u.ac.jp (T.S.); Tel.: +81-52-744-1980 (T.S.)

**Abstract:** Theoretical methods, such as molecular mechanics and molecular dynamics, are very useful in understanding differences in interactions at the single molecule level. In the life sciences, small conformational changes, including substituent modifications, often have a significant impact on function in vivo. Changes in binding interactions between nucleic acid molecules and binding proteins are a prime example. In this study, we propose a strategy to predict the complex structure of DNA-binding proteins with arbitrary DNA and analyze the differences in their interactions. We tested the utility of our strategy using the anticancer drug trifluoro-thymidine (FTD), which exerts its pharmacological effect by incorporation into DNA, and confirmed that the binding affinity of the BCL-2-associated X sequence to the p53 tetramer is increased by FTD incorporation. On the contrary, in p53-binding sequences extracted from FTD-resistant cells, the binding affinity of DNA containing FTD was found to be greatly reduced compared to normal DNA. This suggests that thymidine randomly substituted for FTD in resistant cells may acquire resistance by entering a position that inhibits binding to DNA-binding proteins. We believe that this is a versatile procedure that can also take energetics into account and will increase the importance of computational science in the life sciences.

**Keywords:** binding affinity; thermodynamical effect; molecular mechanics; molecular dynamics

## 1. Introduction

In biomolecules, changes in partial structure, such as amino acid mutations in proteins [1,2], methylation or acetylation of nucleic acids [3–5], can significantly alter their molecular functions. For example, even a single amino acid substitution in a protein sequence can significantly change the structure of the protein, and the biological function may differ accordingly [2]. In addition, it has been shown that the presence or absence of methylation in microRNAs can change the binding interaction with dicer proteins, resulting in differences in the binding rate to complementary RNAs [5]. Therefore, it is very important to observe subtle differences in structure at the single molecule level in biomolecules and to understand their functions. In recent years, it has become possible to experimentally observe subtle differences within a single molecule by using tunneling current-based sequencing [6,7] and other techniques [8]. On the contrary, analysis based on theoretical calculations can predict how changes in substructure will affect intermolecular interactions before they are observed experimentally. Typical simulation calculations based on theoretical physics and chemistry, which are widely used in the life sciences, fall into three main categories. The three methods are the molecular docking method [9,10], which is used to search for bonding conformations between biomolecules; the molecular dynamics (MD) method [11,12], which is used to understand the thermodynamic behavior of molecules; and the molecular orbital (MO) method [13], which can precisely interpret molecular reactivity and intermolecular interactions at the electronic level. By effectively

combining these methods, it has become possible to understand various in vivo environments. In particular, theoretical simulations are effective because they allow detailed analysis of how small conformational changes at the single-molecule level cause changes in interactions with the molecule of interest. However, it is extremely difficult to predict from scratch how the molecular interactions between DNA and its binding proteins, which have undergone minute structural changes, will change compared to the original nucleic acid, using current theoretical calculations. This is because it is difficult to predict the correct binding conformation of a protein that recognizes the double helical DNA structure by the conventional molecular docking method alone, because most of the interfaces that come into contact with DNA are deoxyribose portions. Therefore, we proposed a new strategy, BC-BEP (Binding Conformation and Binding Energy Prediction), to analyze how subtle conformational differences in nucleic acids alter the binding interaction and binding energy between DNA and binding proteins.

BC-BEP uses DNA-protein complex structure data registered in the Protein data bank (PDB) to predict the complex structure of a DNA-binding protein and its target DNA based on homology at various binding conformations. The predicted complex structure is then used as the initial structure, and MD calculations are performed under conditions that mimic the in vivo environment to sample the thermodynamic behavior. These sampled conformations will be used to enable binding energy estimation. To show whether BC-BEP can explain differences in binding changes at the level of a single nucleic acid molecule incorporated into DNA, we tested it with trifluoro-thymidine (FTD) [14], an anticancer drug known to exert high antitumor effects when incorporated into DNA. The binding prediction strategy proposed in this study is strongly expected to be useful in clarifying in vivo characteristics, such as functional changes due to DNA methylation.

## 2. Methods

### *2.1. Conceptual View of Our Procedure for Predicting the Binding Poses and Energies*

In this paper, we propose BC-BEP, which is a strategy for analyzing the distribution of binding energies in in vivo environments by using multiple-target protein-nucleic acid complex structures to predict complexes with arbitrary sequences of nucleic acids based on their binding similarities, and then using molecular dynamics to sample the thermodynamic effects of the binding interactions. This strategy allows theoretical calculations to predict differences in binding interactions caused by small conformational differences at the substituent level within nucleic acids. In the present calculations, we will take as an example the binding interaction between the p53 protein and DNA. The protein has been reported to play a role as a tumor suppressor [15].

#### 2.1.1. Predicting the Complex of DNA Binding Protein and Each DNA Sequence

To predict the binding conformation of any sequence of DNA to the p53 protein, as shown in Figure 1, we first collected the complex structures of the p53 protein and DNA (including partial binding conformations) from the Protein Data Bank (PDB). There are many crystal structures (including substructures) registered in PDB, and we selected 15 crystal structures (PDB ID: 5MG7 [16], 4MZR [17], etc., as shown in Table 1) that accurately predict the complex structure. It has already been reported in many previous studies that the binding motif sequence of p53 protein is (5′-C(A/T)(T/A)G-3′) [18]. Therefore, the DNA structural parts were superposed to minimize the Root Mean Square Deviation (RMSD) of the backbone structure of the binding motif (ribose and phosphate sites) in each crystal structure data. The same procedure was used for the p53 protein portion, where the atomic coordinates forming the protein backbone were superimposed to minimize the RMSD. After superposition of each substructure, the average coordinates of each atom were calculated to create a representative binding conformation that serves as the initial structure for the estimation. Based on this representative binding conformation, a binding conformation with an arbitrary DNA sequence is created by substituting only DNA bases such that the binding interaction between the two binding motifs is not changed. If the

number of bases between two binding motifs in the DNA sequence for which binding is to be inferred is different from the representative binding conformation, the initial structure is created by overlapping the two motifs such that the RMSD of the atomic coordinates of the two motifs is minimized. As the estimated binding conformation to be created here will be followed by structural relaxation, using the energy minimization method, and conformational sampling, using molecular dynamics with the relaxed structure as the initial structure, even a roughly estimated structure will not cause major problems.

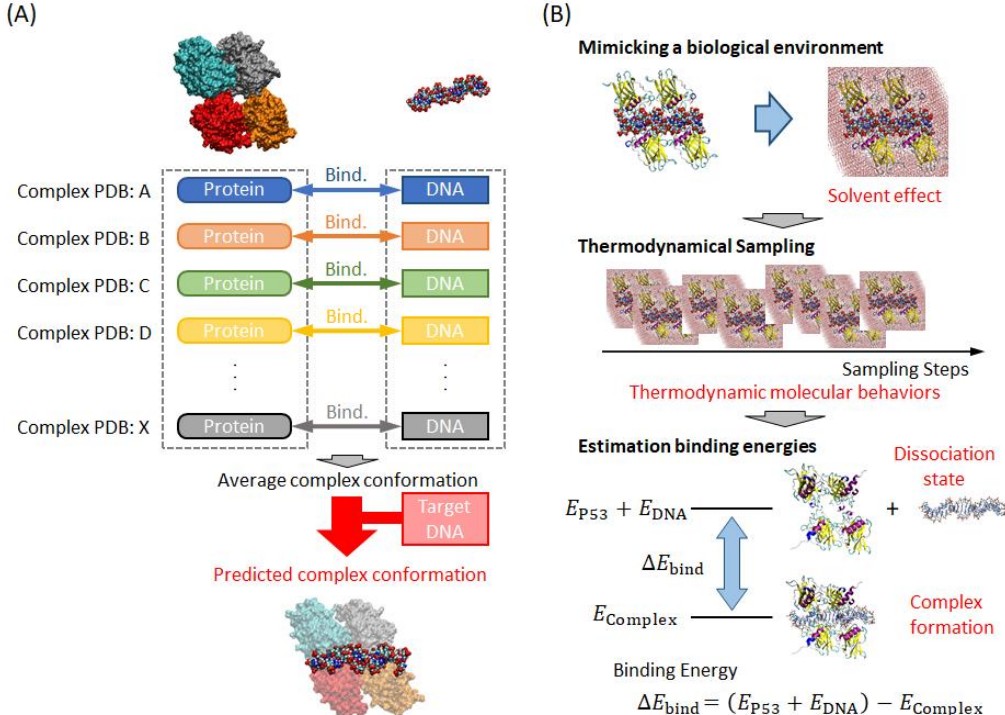

**Figure 1.** Schematic diagram of the proposed theoretical prediction strategy, BC-BEP, of complex conformation and binding affinity. (**A**) After determining the average conformation of the reported Protein Data Bank crystal structures to binding motif sites, the binding structures were predicted based on the sequence similarity of the target DNA. (**B**) Thermodynamic sampling was performed in a mimicked biological environment for the predicted complex structure. Subsequently, the binding energies were estimated to account for the difference in the energy between the dissociated and complex states.

**Table 1.** Crystal structures used to create representative bonding structures.

| PDB ID | Year | X-mer | Title 3 |
|--------|------|-------|---------|
| 5MCT | 2018 | Dimer | Structure 26: 1237–1250.e6, 2018 |
| 5MCU | 2018 | Dimer | Structure 26: 1237–1250.e6, 2018 |
| 5MCV | 2018 | Dimer | Structure 26: 1237–1250.e6, 2018 |
| 5MCW | 2018 | Dimer | Structure 26: 1237–1250.e6, 2018 |
| 5MF7 | 2018 | Dimer | Structure 26: 1237–1250.e6, 2018 |
| 5MG7 | 2018 | Tetramer | Structure 26: 1237–1250.e6, 2018 |
| 6FJ5 | 2018 | Dimer | Structure 26: 1237–1250.e6, 2018 |
| 4MZR | 2014 | Tetramer | J. Mol. Biol. 426: 936–944, 2014 |
| 4HJE | 2013 | Dimer | Nucleic Acids Res. 41: 8368–8376, 2013 |
| 3KMD | 2010 | Dimer | Structure 18: 246–256, 2010 |
| 3KZ8 | 2010 | Dimer | Nat. Struct. Mol. Biol. 17: 423–429, 2010 |
| 2AC0 | 2006 | Tetramer | Mol. Cell 22: 741–753, 2006 |
| 2ADY | 2006 | Tetramer | Mol. Cell 22: 741–753, 2006 |
| 2AHI | 2006 | Tetramer | Mol. Cell 22: 741–753, 2006 |
| 2ATA | 2006 | Dimer | Mol. Cell 22: 741–753, 2006 |

2.1.2. Thermodynamical Structural Sampling and Estimation of Energy Distributions
Energy Minimizations and MD Simulations

Energy minimization was performed using the AMBER 18 program package [19] for each complex structure in ~33,000 water molecules. The AMBER 99 force field [20], general AMBER force field (GAFF) [21] and transferable intermolecular potential with 3 points (TIP3P) force field [22] were used for complex structures and water molecules. We used the GAFF for validation of FTD. Following energy minimization calculations, the MD simulations (canonical ensemble) were performed at 310 K (~37.85 °C) with periodic boundary conditions, using the minimized structure as the initial structure to sample conformations in the biological environment for each structure. The time step was 0.2 fs, and the total simulation time was 10 ns (50,000,000 steps).

Estimation of Binding Energies

We extracted 5000 conformations from each complex structure that was sampled thermodynamically in a mimicked biological environment. We calculated the energies of p53 alone, DNA alone and their complexes by separating the protein and DNA structures from the complex structure. The binding energies were estimated based on the difference in the energy between their complexes from the sum of the energies of the components, according to the following equation:

$$\Delta E_{\text{bind}} = (E_{\text{p53}} + E_{\text{DNA}}) - E_{\text{Complex}} \tag{1}$$

Estimation of binding energy was performed using the Gaussian 16 program package [23]. For validation, we calculated the MM by loading the Amber 99 force field and GAFF into Gaussian 16. To convert the program for energy calculation at the ab initio molecular orbital level, we created a program that discharges in the Gaussian input format.

*2.2. Extraction of DNA from Resistant Cells*

The colon cancer cell line DLD1 and FTD-resistant DLD1 cells were cultured as described in a previous study [24]. DNA was extracted from these cell lines using the QIAamp DNA mini kit (Qiagen, Cat. No. 51304).

**3. Results and Discussion**

*3.1. Prediction of the Complex of p53 Protein with Each DNA Sequence*

Many of the binding interactions of DNA to the p53 protein often occur between the ribose and phosphate groups of DNA and the protein. Therefore, the difference in binding ability between different DNA base species is thought to be largely due to the difference in partial charges caused by the electron-withdrawing and electron-donating properties of the bases. This background makes it very difficult to search for and determine binding conformations from scratch using conventional molecular docking methods. In actuality, Y. Itoh et al. showed that the majority of p53 proteins could not recognize their target sequences in the slide search after non-specific binding to DNA [25]. This experimental result means that the difference in binding energy between the recognition motif and any sequence for p53 protein might be small. On the contrary, our strategy is very efficient because once a representative binding structure is created for one protein, all that remains is base substitution. We have registered the representative binding structures to the p53 protein used here in the PDB format for the Supplement data (PDBs S1 and S3).

To show the usefulness of our strategy, BC-BEP, we selected FTD, which possesses a highly effective anti-tumor potency. The structural difference between FTD and normal thymidine is a substitution with methyl or trifluoromethyl groups (Figure 2a). The combination of FTD and tipiracil hydrochloride has already been approved as a cancer treatment by various regulatory agencies, including the US Food and Drug Administration [14]. FTD has been reported to induce p53-dependent sustained arrest of the cell cycle in the G2 phase, thereby leading to changes in the binding affinity of FTD-incorporated DNA to the p53

protein [4]. From our previous study, it is presumed that thymidine adjacent to adenine and guanine, which has an electron-rich structure, is likely to replace FTD in DNA replication (Figure 2b) [4]. This is because the fluorine atoms in the trifluoromethyl group interact with an electron-rich molecular orbital via halogen bonding. In the validation study, we used normal and FTD-incorporated BCL-2-associated X (BAX) response element sequences binding to p53 (Figure 2c,d). The yellow halftone screening in these figures indicates the area of the p53 recognition motif. Therefore, thymidine in the BAX sequence surrounded by an electron-rich base was substituted with FTD, as shown in Figure 2d. In the second study, we validated p53-binding sequences extracted from cancer cell lines that were resistant to FTDs due to continuous exposure over a long period of time. As shown in Figure 3a, the resistant cells were disrupted and then captured using a consensus-binding sequence for p53. The complementary strand was added based on the captured DNA sequence, as shown in Figure 3b. Based on our previous theoretical predictions, we validated the sequences with FTD substitutions at thymidine positions adjacent to the electron-rich bases (Figure 3c).

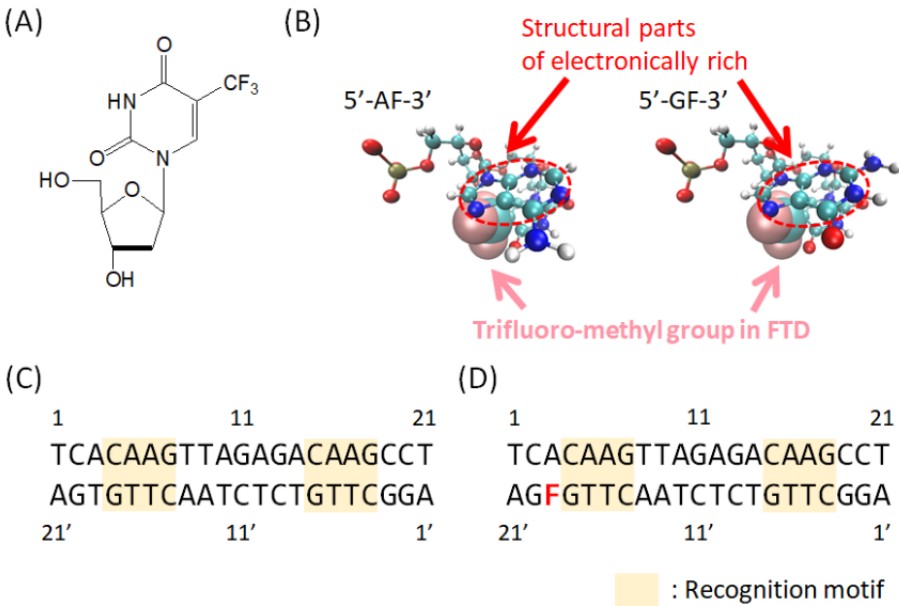

**Figure 2.** Structural features of trifluoro-thymidine (FTD), and test sequences binding to p53. (**A**) The structural formula of FTD and (**B**) examples of adjacent bases that stabilize FTD. The presence of an electron-rich structure in the adjacent bases stabilizes the coordination through an attractive interaction with the trifluoromethyl group of FTD. The BAX response element sequences of (**C**) normal and (**D**) FTD-incorporated DNA.

### 3.2. Thermodynamic Stabilization and Binding Energy Distributions

Based on the predicted binding structures (Supplementary Materials PDBs S1 and S3) of each DNA, i.e., the BAX sequence, and the p53-binding sequence from FTD-resistant cells, we predicted the binding structures (Supplementary Materials PDBs S2 and S4) of the BAX sequence incorporating FTD and the p53-binding sequence from resistant cells incorporating FTD. For these complexes, the conformational structures were sampled using the MD method at a biological temperature (310 K [~37 °C]) after equilibration for a sufficient time. Figure 4a,b shows the RMSDs of the DNA positions calculated for these sampling structures after superposing the p53 protein structures for BAX and binding sequences from FTD-resistant cells, respectively. The variation for a normal DNA and an FTD-incorporated DNA are shown as blue and orange lines, respectively. In the BAX sequence, FTD-incorporated DNA was found to have lower thermal oscillations than normal DNA, while normal DNA from FTD-resistant cells exhibited increased thermal stability compared to FTD-incorporated DNA. Due to these differences in thermal stability,

shown in Figure 5a,b, the difference in the binding energy distributions were fitted to the Gaussian-type function and that of box plots. In these figures, the blue and orange lines show the distribution of a normal and FTD-incorporated DNA, respectively. The p-values using *t*-test between the two distributions are $3.04 \times 10^{-26}$ and ~0, respectively. In these thermal samplings, the average relative distance between each DNA and the p53 is shown in Figure 6. The red circle indicates the position of FTD. Red- and blue-dashed circles indicate the average position of the FTD incorporated and the normal DNA loop position, respectively.

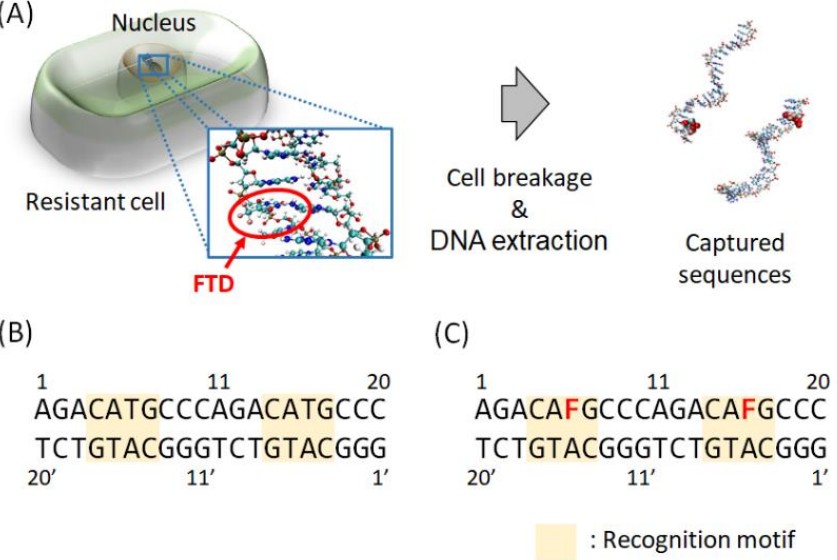

**Figure 3.** Validation using p53-binding DNA extracted from trifluoro-thymidine (FTD)-resistant cells. (**A**) The resistant strains were disrupted and p53-binding sequences were extracted. The sequences of (**B**) normal and (**C**) FTD-incorporated DNA used for verification.

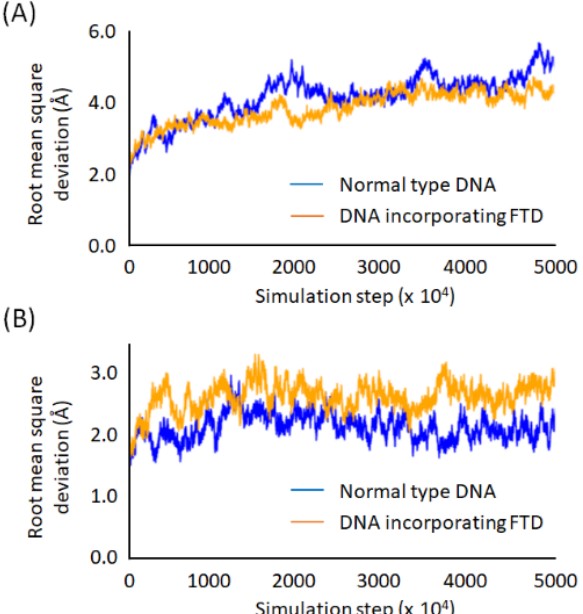

**Figure 4.** Thermal vibration of each DNA in complex structures. The root mean square deviation (RMSD) analyses of the (**A**) BAX response element sequences and (**B**) p53-binding sequences extracted from trifluoro-thymidine (FTD)-resistant cells. The variation for a normal and FTD-incorporated DNA is shown as blue and orange lines, respectively.

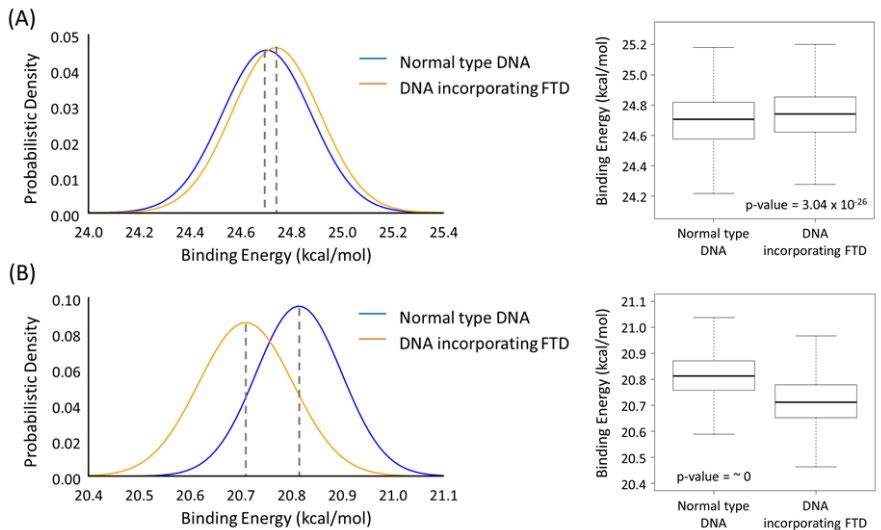

**Figure 5.** Comparison of binding energies. The distributions of p53-binding energy fitted to the Gaussian-type function and box plots for the (**A**) BAX response element sequences and (**B**) p53-binding sequences extracted from trifluoro-thymidine (FTD)-resistant cells. The blue and orange lines indicate the distribution for a normal and FTD-incorporated DNA, respectively. The p-values between the two distributions are (**A**) $3.04 \times 10^{-26}$ and (**B**) ~0.00, respectively.

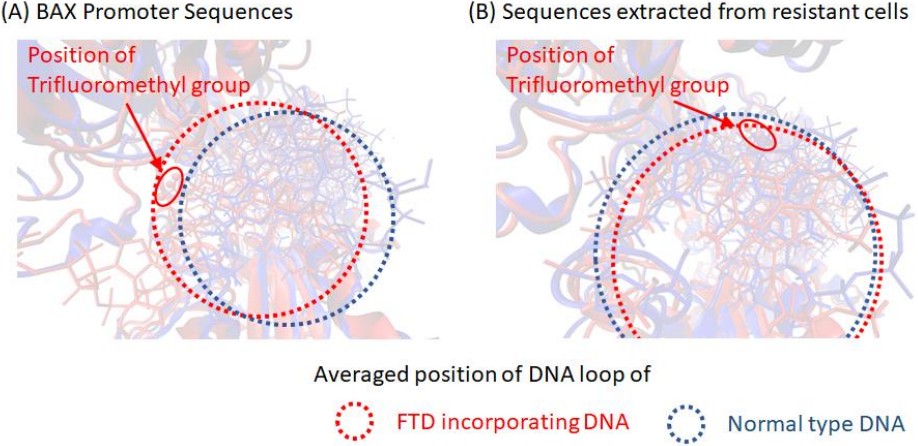

**Figure 6.** Comparison of the positions of proteins and each DNA. Representative superposition of binding conformations of the (**A**) BAX promoter sequences and (**B**) sequences extracted from FTD-resistant cells to p53 protein. The red circle indicates the position of trifluoro-thymidine (FTD). Red-dashed and blue-dashed circles indicate the average position of the FTD-incorporated and normal DNA loop, respectively.

In our study, the binding energies were calculated using MM with the Amber force field, which represents qualitative rather than quantitative properties. As the probability of DNA recognition by p53 has been reported to be very low [25], the difference in the binding energy caused by sequence differences is also expected to be insignificant. Moreover, we found a slight increase in binding energy caused by FTD-incorporated DNA. However, this indicates that FTD-incorporated DNA binds more strongly to DNA-binding proteins and has higher recognition than normal DNA. In contrast, in p53-binding sequences extracted from FTD-resistant cells, we found that the binding affinity of FTD-incorporated DNA was greatly reduced compared to that of normal DNA. Therefore, we suggest that the random substitution of thymidine in FTD-resistant cells might confer resistance to FTDs by occupying a position that prevents them from binding to DNA-binding proteins. Corroborating this result, we observed that the average relative distance between FTD-incorporated DNA and p53 protein was closer for the BAX promoter sequence than for

the normal DNA, whereas for the sequence extracted from the resistant strain, the FTD-incorporated sequence tended to be farther away (Figure 6). This method can be used to calculate the binding energy for any mutation containing p53. In addition to p53, this method also facilitates the analysis of the structural and energetic properties of other DNA-binding molecules, such as NANOG. Furthermore, although the energy distributions were estimated using the MM method, future studies should analyze the intermolecular interactions and estimate the binding energies using quantum mechanical methods as needed. Therefore, this strategy has several applications and great potential for expansion by modifying the computer specifications.

## 4. Conclusions

Our study established a novel strategy for predicting p53-binding energies based on a conformational prediction of the arbitrary DNA complex to p53, using theoretical methods, such as MM and MD. Validation studies using an anti-tumor drug, FTD, that exerts its pharmacological effects by incorporating into DNA, exhibited an increased binding affinity for the p53 tetramer. As estimated from previous studies, the increase in binding energy was not significant, even in the presence of increased p53 function. In contrast, the FTD-resistant cell line revealed a substantial decrease in binding energy. Thus, the proposed strategy enabled the prediction of the functional response of p53 that cannot be predicted by experiments that consider the difference in the binding energy between any DNA sequence and a control sequence. As the proposed method was designed as a general-purpose method, it can be extended to other DNA-binding proteins, and we believe that it is a method that can be used to discuss the activation status of biological functions, including cell proliferation and inhibition in vivo. We have already applied this method to the NANOG protein and predicted that the binding strength changes between methylated and unmethylated DNA, and we have also been able to experimentally and theoretically explain the functional changes between the two cases; the paper is currently under submission. Therefore, we believe that this proposed method is highly versatile and applicable for calculating the binding energies of not only p53 protein, but also other DNA-binding proteins and biomolecules, and it highlights the importance of computational science in life sciences.

**Supplementary Materials:** The following supporting information can be downloaded at: https://www.mdpi.com/article/10.3390/app13010510/s1, PDB S1: BAX-normal type DNA; PDB S2: BAX-DNA incorporating FTD; PDB S3: Resistant Cell-normal type DNA; PDB S4: Resistant Cell-DNA incorporating FTD.

**Author Contributions:** J.K. designed the analyses and performed all computational analyses. J.K., H.H. and T.S. performed identifying and gathering information on issues related to p53. M.K. performed some experiments. J.K., H.H., M.K. and T.S. discussed this research in detail. J.K. and T.S. wrote the manuscript. All authors have read and agreed to the published version of the manuscript.

**Funding:** This work was supported by grants from grants-in-aid of research by companies; Osaka Cancer Society (J.K.) and Takeda Science Foundation (J.K.), and from KAKENHI for scientific research (B) (J.K., grant number 22H03686).

**Institutional Review Board Statement:** Not applicable.

**Informed Consent Statement:** Not applicable.

**Data Availability Statement:** Data is contained within the article or supplementary material.

**Conflicts of Interest:** The authors declare no conflict of interest.

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
