# Peer review of "Theoretical Computational Analysis Predicts Interaction Changes Due to Differences of a Single Molecule in DNA"

_applsci, doi:10.3390/app13010510_

Round 1

Reviewer 1 Report

This paper uses computational methods to study intermolecular interactions. The overall idea is reasonable, but there are some issues worth noting before publication:

1. The title of the paper is too broad, it is recommended to focus on the research system.

2. I feel that the calculation of binding energy is not clearly described. For example, when calculating the binding energy between p53 and DNA, is the energy of p53 or DNA calculated separately the potential energy at 0k? Are there multiple situations for the spatial configuration of the Complex? Is the configuration of p53 or DNA in the Complex the same as when calculating the two separately? If they are different, is it possible that the contribution of binding energy is partly due to the difference in configuration?

Author Response

First, we appreciate Reviewer for refereeing our manuscript and giving comments.

1) The title of the paper is too broad, it is recommended to focus on the research system.

As mentioned by referee, we felt that the title was too broad, so we have changed it as follows. Thank you.

“Theoretical computational analysis predicts interaction changes due to differences of a single molecule differences in DNA”

2) I feel that the calculation of binding energy is not clearly described. For example, when calculating the binding energy between p53 and DNA, is the energy of p53 or DNA calculated separately the potential energy at 0k? Are there multiple situations for the spatial configuration of the Complex? Is the configuration of p53 or DNA in the Complex the same as when calculating the two separately? If they are different, is it possible that the contribution of binding energy is partly due to the difference in configuration?

Thank you for pointing this out. As the reviewer pointed out, I think the wording was a bit difficult to convey. First of all, this calculation does not take into account the zero-point vibrational energy, since the energy is calculated at the MM level. To simplify the calculation, the molecular motion sampling by MD is used to perform structural sampling by complex, from which the protein and DNA are separated to obtain the distribution of interaction energies. In light of the above, the Method section was modified as follows.

“We extracted 5,000 conformations from each complex structure that was sampled thermodynamically in a mimicked biological environment. We calculated the energies of p53 alone, DNA alone, and their complexes, by separating the protein and DNA structures from the complex structure. The binding energies were estimated based on the difference in the energy between their complexes from the sum of the energies of the components according to the following equation:”

Reviewer 2 Report

This article by Jun Koseki, " Theoretical computational analysis predicts single molecule 2 differences". This is a useful attempt and will be helpful reference for many researchers. The work has scientific relevance, presents expressive results. Overall Manuscript is good and well design.

Minor suggestion.

Authors are suggested to replace in vivo with in-vivo or in vivo

Author Response

First, we appreciate Reviewer for refereeing our manuscript and giving comments.

1) Authors are suggested to replace in vivo with in-vivo or in vivo.

Thank you for pointing out. We have changed all "in vivo" to "in vivo".

Reviewer 3 Report

The authors of this paper have developed a new strategy to predict the complex structure of DNA-binding proteins using binding conformation and binding energy prediction (BC-BEP).

The paper is very well written and is suitable for publication in Applied Sciences. I have a few minor comments for the authors.

1)      The authors should mention the full form of acronyms in the abstract. Not everyone will know what BC-BEP, BAX and FTD means.

 2) Typo in line 58. It should be DNA

3) Why did the author choose p52 protein? They mention that it plays a role in tumour suppressor, but dont give any reference.

Author Response

First, we appreciate Reviewer for refereeing our manuscript and giving comments.

1) The authors should mention the full form of acronyms in the abstract. Not everyone will know what BC-BEP, BAX and FTD means.

Thank you very much. We think the referee is right. Therefore, BAX is written as BCL-2-associated X, and FTD is also written with trifluoro-thymidine (FTD). BC-BEP is the name of the strategy named by the authors in this study, so we have removed the abbreviation in the abstract.

2) Typo in line 58. It should be DNA.

As mentioned by referee, there was a spelling error and we have corrected it from "DAN" to "DNA". Thank you.

3) Why did the author choose p52 protein? They mention that it plays a role in tumour suppressor, but don’t give any reference.

Thank you for pointing this out. I have cited the following article on p53 and cancer for the reader's broad understanding.

  1. Kanapathipillai M. Treating p53 Mutant Aggregation-Associated Cancer. Cancers (Basel). 2018 May 23;10(6):154. doi: 10.3390/cancers10060154. PMID: 29789497; PMCID: PMC6025594.